# Enhanced Stability of Solution-Processed Indium–Zinc–Tin–Oxide Transistors by Tantalum Cation Doping

Haiyang Xu [1], Pingping Li [2,3], Zihui Chen [4], Bing Yang [4], Bin Wei [2,3], Chaoying Fu [5,*], Xingwei Ding [2,3,*] and Jianhua Zhang [2]

1 School of Materials Science and Engineering, Shanghai University, Shanghai 200072, China; hyxu@shu.edu.cn
2 Key Laboratory of Advanced Display and System Application, Ministry of Education, Shanghai University, Shanghai 200072, China; 1270886521@shu.edu.cn (P.L.); bwei@shu.edu.cn (B.W.); jhzhang@oa.shu.edu.cn (J.Z.)
3 School of Mechatronics and Automation, Shanghai University, Shanghai 200072, China
4 School of Microelectronics, Shanghai University, Shanghai 200072, China; c546923746@shu.edu.cn (Z.C.); byang@shu.edu.cn (B.Y.)
5 Huzhou Key Laboratory of Medical and Environmental Applications Technologies, School of Life Sciences, Huzhou University, Huzhou 313000, China
* Correspondence: 03064@zjhu.edu.cn (C.F.); xwding@shu.edu.cn (X.D.); Tel.: +86-21-56331976 (X.D.); Fax: +86-21-39988216 (X.D.)

**Abstract:** Highly stable metal oxide thin film transistors (TFTs) are required in high-resolution displays and sensors. Here, we adopt a tantalum cation ($Ta^{5+}$) doping method to improve the stability of zinc–tin–oxide (ZnSnO) TFTs. The results show that $Ta^{5+}$-doped TaZnSnO TFT with 1 mol% concentration exhibits excellent stability. Compared with the undoped device, the oxygen vacancy defects of TaZnSnO thin films reduce from 38.05% to 18.70%, and the threshold voltage shift ($\Delta V_{th}$) reduces from 2.36 to 0.71 V under positive bias stress. We attribute the improved stability to the effective suppression of the oxygen vacancy defects, which is confirmed by the XPS results. In addition, we also prepared TaInZnSnO TFT devices with 1 mol% $Ta^{5+}$ doping concentration. Compared with the 1 mol% $Ta^{5+}$-doped TaZnSnO TFTs, the $\mu$ increases two-fold from 0.12 to 0.24 $cm^2/Vs$, and the $V_{th}$ decreases from 2.29 to 0.76 V in 1 mol% $Ta^{5+}$-doped TaInZnSnO TFT with an In:Zn:Sn ratio of 4:4:3, while the device remains highly stable with a $\Delta V_{th}$ of only 0.90 V. The injection of $Ta^{5+}$ provides a novel strategy for the enhancement of the stability in ZnSnO-based TFTs.

**Keywords:** thin film transistors; stability; doping; mobility; threshold voltage

## 1. Introduction

Amorphous oxide semiconductor (AOS) thin film transistors (TFTs) are widely applied due to their higher mobility and better large-area uniformity than amorphous silicon TFTs. For example, they are used in displays, CMOS image sensors, and touch sensors [1,2]. Today, AOS TFTs can be fabricated by vacuum deposition processes such as magnetron sputtering, atomic layer deposition (ALD), and chemical vapor deposition. However, the high fabrication cost of these vacuum deposition processes limits their large-scale application. As a newly developed AOS fabrication method, solution processing techniques offer the possibility of depositing thin films in a simple printing or coating manner, enabling the fabrication of low-cost and high-performance electrical devices [3–5], and the chemical composition of AOS films is easy to control compared to other physical deposition methods [6–8].

Solution-processed zinc–tin–oxide (ZnSnO) TFTs were widely investigated due to their excellent properties, such as a low-temperature synthesis process, simple fabrication, low cost, and resource sustainable ability [9–11]. However, oxygen vacancies have some negative effects on oxide films. The situation of oxygen vacancy is complicated because the vacancies are in several charge states. For both binary and complex oxides, in contrast to

alkali halides, oxygen vacancy in ionic oxides can have two charge states: the one-electron $F^+$ center and the two-electron $F$ center. The absorption bands of the $F^+$ and $F$ centers in most oxides occur at different energies. In addition, the diffusion characteristics of vacancies depend on their charge state [12]. Owing to the oxygen vacancy defects, it is difficult to control the mobility, threshold voltage, stability, etc., in ZnSnO TFTs [13]. It is known that these problems can be solved by doping appropriate cations as suppressors of oxygen vacancy. The principle is to control the oxygen vacancy-related trap states by introducing an element that has higher binding energy with oxygen [14]. In recent years, several doping elements, including Zr [6], Ba [2], P [15], and Sb [16], have been reported to improve the stability of devices by suppressing oxygen vacancy. Unfortunately, they both have large threshold voltages ($V_{th}$) (~5−10 V) [6,16–18]. Therefore, one of the keys is to reduce the $V_{th}$ while improving stability. Tantalum ($Ta^{5+}$) has a larger oxygen bonding dissociation energy (Ta-O dissociation energy of 799.1 kJ/mol) than those of Zn (Zn-O dissociation energy of 159.4 kJ/mol) and Sn (Sn-O dissociation energy of 531.8 kJ/mol) [19], which is expected to be an effective oxygen vacancy inhibitor. However, $Ta^{5+}$ doping may cause a decrease in field-effect mobility ($\mu$) and an increase in $V_{th}$. Indium ions ($In^{3+}$), possessing a special electron configuration of $(n-1) d^{10}ns^0$ ($n$ is the principal quantum number) [20], play a significant role in improving the $\mu$ of the TFTs. However, the effect of $In^{3+}$ on the performance of TaZnSnO TFTs still needs to be further investigated.

Dielectric layers are also critical to TFT device performance, and high-quality dielectric layers can be prepared via ALD, which uses a binary reaction split into two self-limiting chemical reactions in a repeated alternate deposition sequence. The ALD technique has a lot of advantages, including high controllability for composition and thickness, excellent reproducibility, and low deposition temperatures [21,22]. $Al_2O_3$, with good insulating properties grown by ALD (ALD-$Al_2O_3$), is one of the most widely studied materials [23]. Until now, the solution-processed method has not been investigated to prepare films on ALD-$Al_2O_3$, due to the low hydrophilicity of the ALD-$Al_2O_3$ surface [24]. To overcome this problem, we adopted UV–ozone treatment to improve the hydrophilicity of the ALD-$Al_2O_3$ surface.

Herein, we first report the TaZnSnO and TaInZnSnO TFTs fabricated by the solution method on ALD-$Al_2O_3$. The effect of $Ta^{5+}$ doping on the properties of TaZnSnO thin films and their associated application to TFTs are investigated. $Ta^{5+}$ doping can improve the stability of the ZnSnO thin films and decrease carrier concentration due to the suppression of oxygen vacancies. Moreover, $Ta^{5+}$ injection was also developed to explore the electrical properties and stability of their associated TaInZnSnO TFTs.

## 2. Experiments

The 0.3 M ZnSnO precursor was synthesized by dissolving $Zn(CH_3COO)_2 \cdot 2H_2O$ and $SnCl_2 \cdot 2H_2O$ in 2-methoxyethanol with the molar ratio of Zn:Sn = 4:3 in an airtight glove box to cut off $O_2$ and $H_2O$, and the water and oxygen content of the glove box in the laboratory are lower than 0.1 ppm and lower than 10.7 ppm, respectively. After the precursor solution was stirred for 15 min to fully dissolve the solute, ethanolamine was added as a stabilizer to avoid turbidity and precipitation of the precursor solution. $TaCl_5$ was then dissolved into the ZnSnO solution to form the TaZnSnO precursor solutions with $Ta^{5+}$ molar ratios of 0.5, 1, and 2 mol%. We also prepared 0.3 M TaInZnSnO precursor by dissolving $In(NO_3)_3 \cdot xH_2O$ into the 1 mol% $Ta^{5+}$ doping concentration TaZnSnO precursor solution with the molar ratio of In:Zn:Sn = 4:4:3. Then, the prepared precursor solution was stirred in a water bath heating pot with a magnetic stirrer at 60 °C for 2 h, followed by stirring for 12 h to process homogeneous hydrolysis.

Before coating, the $Al_2O_3$ gate dielectrics of approximately 50 nm thick were formed on the heavily doped *p*-type single crystal semiconductor Si (100) substrates (1–10 Ω·cm) using ALD (TFS-200, Beneq, Finland) at 250 °C with trimethylaluminum and $H_2O$, and the purge time was 10 s. Then, the hydrophilicity of the $Al_2O_3$ gate dielectrics was improved by UV–ozone treatment for 10 min to enhance the uniformity of the coating. After that, the

precursor solutions were spin-coated on the $Al_2O_3$ films at 500 r/min for 5 s, followed by 2000 r/min for 30 s. Then, they were heated on a hot plate at 150 °C for 15 min. Finally, the devices were annealed at 500 °C for 1 h. Al 200-nm-thick was deposited by thermal evaporation to form the source and drain electrodes through a shadow mask; the channel width (*W*) and channel length (*L*) were 1000 and 50 μm, respectively. The schematic structure of the TFTs with various doping ratios is illustrated in Figure 1a. The prepared devices were annealed at 300 °C in the air for 15 min on the hot annealing furnace. This step is for the formation of good ohmic contact between the semiconductor and metal electrodes to reduce the resistivity. To evaluate the optical performance of thin films, ZnSnO and TaZnSnO films were grown on quartz glass substrates through the same process.

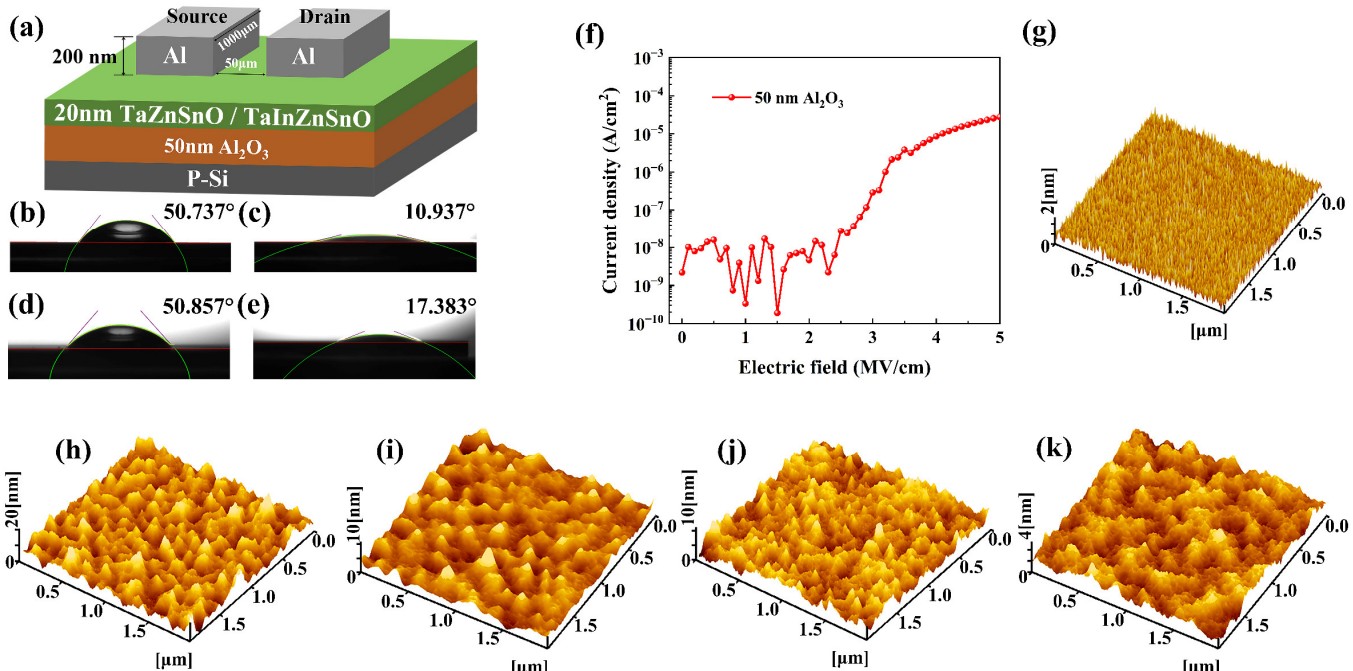

**Figure 1.** (**a**) The schematic structure of the TaZnSnO/TaInZnSnO TFTs. The water contact angles of $Al_2O_3$ films before (**b**) and after (**c**) UV–ozone treatment, and the contact angles of $Al_2O_3$ before (**d**) and after (**e**) UV–ozone treatment using 2-methoxyethanol. (**f**) Leakage characteristic curve of $Al_2O_3$ film. AFM images of (**g**) 50 nm $Al_2O_3$ dielectric layer ($2 \times 2$ μm$^2$) and TaZnSnO thin films with different $Ta^{5+}$ doping contents (**h**) 0, (**i**) 0.5, (**j**) 1, and (**k**) 2 mol%.

The thermal decomposition characteristics of the precursor solution were obtained by thermogravimetric analysis (TGA). The test conditions were varied from room temperature to 800 °C with a 10 °C/min heating rate in an air environment. The water contact angle test was performed by using contact angle meter (Contact Angle Meter Model SL200KS, Solon, China). The surface morphologies of the films were characterized by atomic force microscope (AFM, nanonaviSPA-400 SPM, SII Nano Technology Inc., Chiba City, Japan). Measurement of optical transmittance was performed using spectrometer (HITACHI U-3900H, Tokyo, Japan) at room temperature. The crystal structure of the thin films was characterized by X-ray diffraction (XRD, Rigaku D/max-rB, Tokyo, Japan), and the composition was analyzed by X-ray photoelectron spectroscopy (XPS, Thermo Scientific K-Alpha+, Waltham, MA, USA). Electrical characteristics were tested with a semiconductor characterization system (Keithley, 4200-SCS, Cleveland, OH, USA).

## 3. Results and Discussion

In order to improve the hydrophilicity of the $Al_2O_3$ film and facilitate the subsequent spin coating, we performed UV–ozone treatment on it, and used the contact angle (*θ*) and

surface free energy ($\gamma$) to characterize the difference in the hydrophilicity of the film. The surface free energy of thin film can be calculated with the following formula [25]:

$$\gamma_S = \gamma_S^d + \gamma_S^p \tag{1}$$

$$\gamma_L(1 + \cos\theta) = 2\sqrt{\gamma_S^d\gamma_L^d} + 2\sqrt{\gamma_S^p\gamma_L^p} \tag{2}$$

where $\gamma_S$ and $\gamma_L$ are the surface free energy of solid and liquid, and the *p* component and *d* component correspond to their polar component and dispersion component, respectively. The water contact angles of $Al_2O_3$ films before and after UV–ozone treatment are 50.737° and 10.937°, as shown in Figure 1b,c. In addition, we measured the contact angles of $Al_2O_3$ before and after UV–ozone treatment using 2-methoxyethanol, as shown in Figure 1d,e, and calculated the corresponding $\gamma_S$. The contact angle (17.383°) of the $Al_2O_3$ surface of 2-methoxyethanol after UV–ozone treatment is also smaller than the contact angle 50.857° of the surface before UV–ozone treatment. According to the contact angle test results of the film, the $\gamma_S$ of the $Al_2O_3$ film after UV–ozone treatment is 89.794 mN/m, and the $\gamma_S$ before UV–ozone treatment is 59.285 mN/m. The lower water contact angle and higher $\gamma_S$ indicate that the film is more hydrophilic. The results show that UV–ozone treatment can significantly improve the hydrophilicity of $Al_2O_3$ films, which makes a better condition for the subsequent spin coating of the ZnSnO-based films. The leakage current characteristics of 50 nm ALD-$Al_2O_3$ dielectric are charted in Figure 1f. The $Al_2O_3$ produces a leakage current density of $10^{-8}$ A/cm$^2$ at 2 MV/cm, which is suitable for TFT application. The surface morphologies of the $Al_2O_3$ were analyzed by AFM, as shown in Figure 1g. The $Al_2O_3$ thin film exhibits a smooth surface with a root mean squared (RMS) value of 0.33 nm. The smooth surface of the gate dielectric can suppress the surface roughness-induced leakage current and achieve expeditious carrier transport in the channel. Figure 1h–k present the surface morphologies of the TaZnSnO thin films with different Ta$^{5+}$ doping concentrations. The RMS values are 2.43, 1.11, 1.30, and 0.66 nm for TaZnSnO thin films with Ta$^{5+}$ doping concentrations of 0, 0.5, 1, and 2 mol%, respectively. The RMS roughness of the films decreases slightly with the increase in doping concentration. Rough surface morphology of the undoped ZnSnO thin films is associated with the growth of the columnar structure; the growth of the columnar structure in TaZnSnO films is suppressed by doped Ta element, which may result from the formation of stresses by the different ion sizes of Zn and Sn, and the segregation of Ta cations [26]. The smooth surface roughness of the active layer facilitates the adhesion ability to the source and drain electrodes.

    Figure 2a shows the GIXRD pattern of TaZnSnO films with different Ta$^{5+}$ doping ratios. No XRD peak corresponding to TaZnSnO films is found, which indicates that the TaZnSnO films have an amorphous structure. Grain boundaries usually act as preferential paths for impurity diffusion and leakage current, resulting in an inferior dielectric performance. In comparison, an amorphous structure provides a fast transport path for carriers [27,28], which may be beneficial for the preparation of large-size oxide TFTs. As shown in Figure 2b, two apparent weight loss stages could be observed for all the solutions by TGA analyses. The initial weight loss stage happened in the range of 70–150 °C. This phenomenon was caused by the evaporation of volatile nitrates and the hydrolysis of $Zn(CH_3COO)_2 \cdot 2H_2O$, $SnCl_2 \cdot 2H_2O$, and $TaCl_5$. In the temperature range of 150–500 °C, only a slight loss of weight due to the conversion of the relevant hydroxides to the corresponding oxides through dehydroxylation and condensation reactions. No apparent weight loss was observed at temperatures above 500 °C, implying that the precursor solution completely formed a dense metal oxide film. Therefore, we chose 500 °C as the annealing temperature of the thin film according to the analysis results.

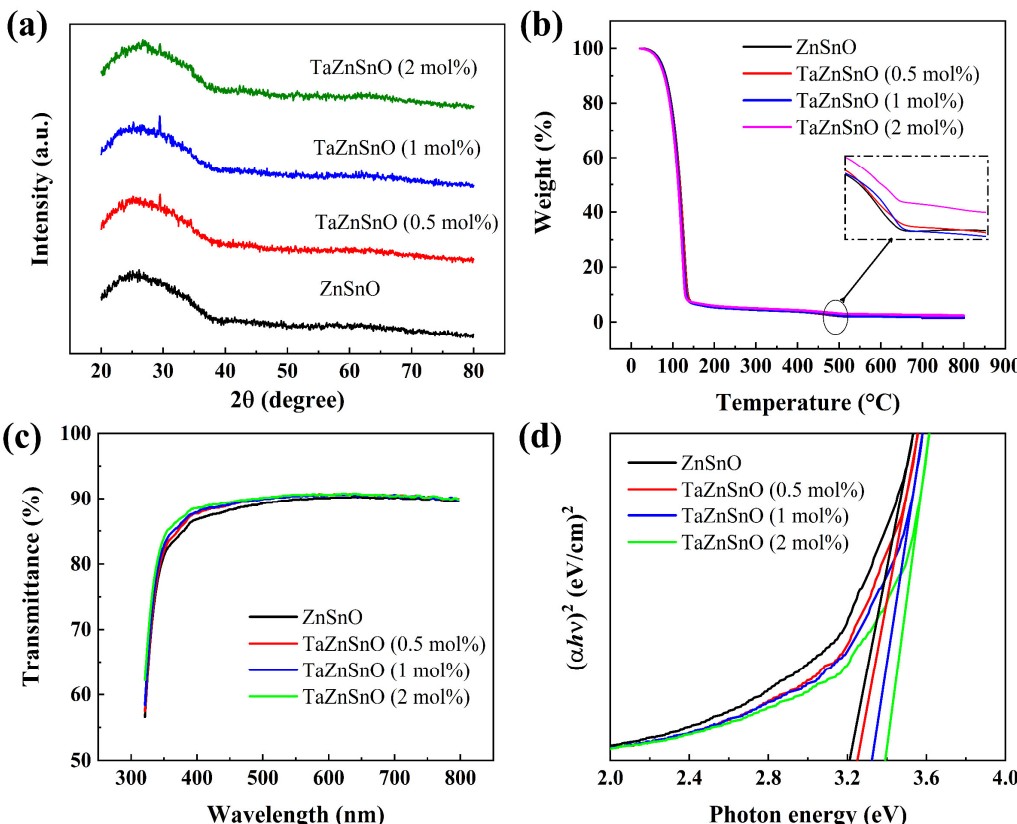

**Figure 2.** (**a**) GIXRD pattern, (**b**) TGA curve, (**c**) optical transmittance spectra, and (**d**) optical bandgap of TaZnSnO thin films with different $Ta^{5+}$ doping concentrations.

The optical transmittance spectra of the TaZnSnO films were analyzed to depict the effect of $Ta^{5+}$ doping on the optical transmittance, as shown in Figure 2c. The optical transmittance of all samples shows about 90% transmission in the visible region. With the increase in $Ta^{5+}$ content, the average transmittance of TaZnSnO films increases slightly, indicating that TaZnSnO film has great potential to be applied on transparent electronic devices. The transmittance became slightly higher by increasing $Ta^{5+}$ addition, due to its wider bandgap. The optical bandgap ($E_g$) is related to the absorption coefficient ($\alpha$) by the following equation [29]:

$$\alpha h\nu = \left(h\nu - E_g\right)^{\frac{1}{2}} \tag{3}$$

where $\alpha$, $h$, and $\nu$ are the optical absorption coefficient, Planck's constant, and the incident photon frequency, respectively. The absorption coefficient ($\alpha$) can be extracted by the following equation:

$$\alpha = \left(\frac{1}{d}\right)\ln\left(\frac{1}{T}\right) \tag{4}$$

where $T$ is the transmittance and $d$ is the film thickness. Figure 2d shows a plot of $(\alpha h\nu)^2$ versus photon energy. The $E_g$ of 0, 0.5, 1, and 2 mol% $Ta^{5+}$-doped ZnSnO films are 3.21, 3.25, 3.32, and 3.39 eV, respectively, which are determined from the absorption spectra by extrapolation. However, there are some defects that may have some of their energy levels in the band gap. We use the Urbach rule to fit the band gap as an "estimation" only, but not the true "calculation" [30]. A larger $E_g$ requires the carriers to have higher energy to shift from the valence band toward the conduction band. The generation of carriers by AOS depends on the generation of oxygen vacancies according to the following equation [6]:

$$O_O^X \leftrightarrow V_{\ddot{O}} + 2e^- + \frac{1}{2}O_2 \tag{5}$$

Hence, $Ta^{5+}$ doping reduces the carrier concentration and reduces the defect states generated by oxygen vacancies.

To further investigate the effect of $Ta^{5+}$ doping on the chemical bonding states of TaZnSnO thin films, XPS was used to test the O1s spectra of the TaZnSnO thin films with different $Ta^{5+}$ doping concentrations; Figure 3 shows the XPS survey spectra of the TaZnSnO thin films, and the XPS peaks of the major lattice components of Zn (2s, 2p1, and 2p3), Sn (3s, 3p1, 3p3, 3d3, 3d5, 4s, 4p, and 4d), Ta (4f), O (1s), C (1s), and Cl (2p) are visible. In addition, the photoelectron peaks of the main elements, Zn, O, Auger Zn LMM, and O KLL, can be seen, and it is almost consistent with previously reported data [31].

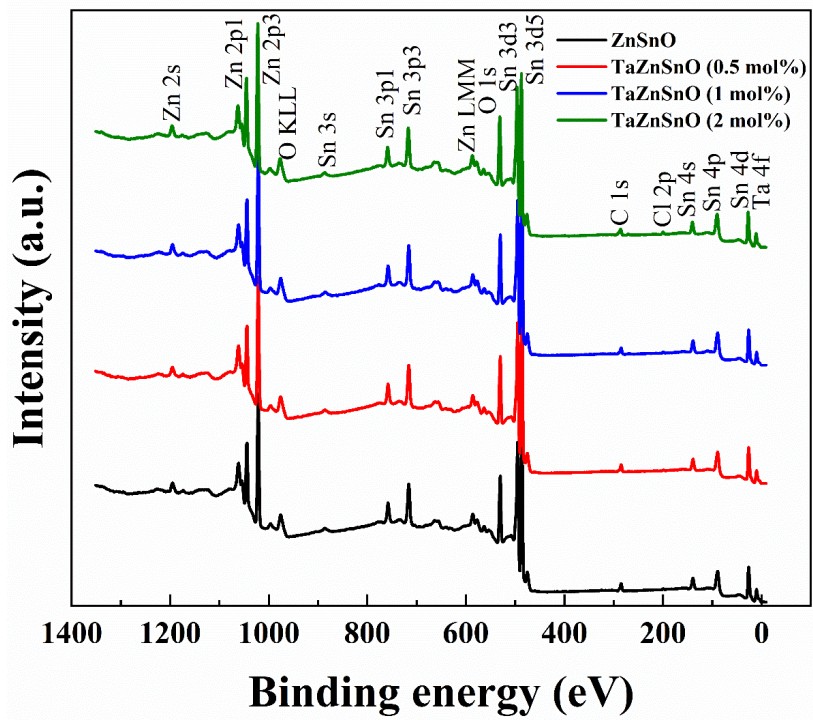

**Figure 3.** XPS survey spectra of the TaZnSnO thin films with different $Ta^{5+}$ doping.

As shown in Figure 4a–d, all peaks were calibrated by taking the carbon C1s (284.5 eV). The XPS O1s spectra of the TaZnSnO films were fitted by three component peaks with the Gaussian distribution and approximate center frequency of 529.8 eV ($O_1$), 530.5 eV ($O_2$), and 531.6 eV ($O_3$), respectively. The $O_1$ is attributed to the bonding of the oxygen ion ($O^{2-}$) to metal ions (M–O) such as $Ta^{5+}$, $Zn^{2+}$, and $Sn^{2+}$ ($Sn^{4+}$). The $O_2$ is related to the oxygen-related defect states in TaZnSnO films [32]. The $O_3$ is associated with loosely bound oxygen on the surface of films such as C, $O_2$, $O_2$, and $H_2O$ [33]. $O_2/(O_1 + O_2 + O_3)$ is defined as the relative quantity of oxygen vacancy in the TaZnSnO thin films. The ratio of $O_2/(O_1 + O_2 + O_3)$ was calculated to be 38.05%, 26.26%, 18.70%, and 24.81% for TaZnSnO films with $Ta^{5+}$ doping concentrations of 0, 0.5, 1, and 2 mol%, respectively. The oxygen vacancy concentration decreased to a minimum with $Ta^{5+}$ content in 1 mol% for the TaZnSnO film, due to the greater difference in electronegativity between tantalum (1.50) [34] and O (3.44) than that between Zn (1.65), Sn (1.96), and O [35]. Therefore, $Ta^{5+}$ has a stronger attraction ability to oxygen ions. Moreover, the ratio of oxygen vacancy defects increased from 18.70% to 24.81%, with the doping concentration of $Ta^{5+}$ increasing from 1 mol% to 2 mol%, which may be caused by over doping leading to more trap states and the result concurring with the *SS* (Table 1). So, we can conclude that the appropriate amount of $Ta^{5+}$ doping can effectively suppress the oxygen defect concentration in the ZnSnO film.

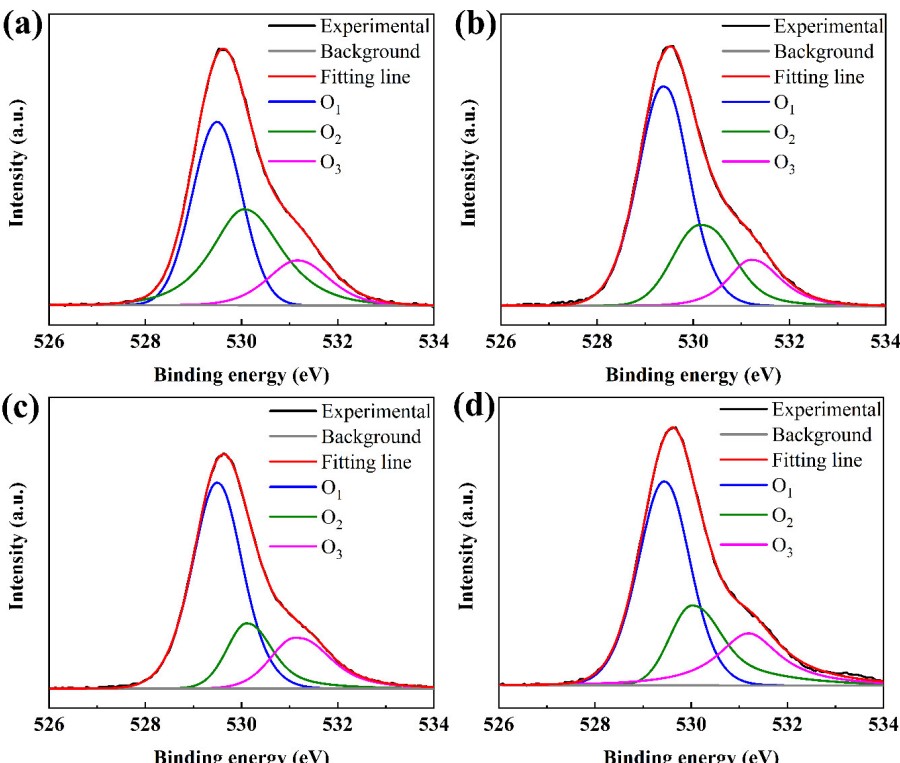

**Figure 4.** XPS of O1s spectra of the TaZnSnO thin films with different $Ta^{5+}$ doping levels of (**a**) 0, (**b**) 0.5, (**c**) 1, and (**d**) 2 mol%.

**Table 1.** Electrical performance of the TaZnSnO TFTs with different $Ta^{5+}$ doping concentrations.

| Device | $V_{th}$ (V) | $\mu$ (cm²/Vs) | $I_{on}/I_{off}$ | SS (V/Decade) |
|---|---|---|---|---|
| ZnSnO | 0.29 ± 0.03 | 0.38 ± 0.080 | ~$10^6$ | 0.103 |
| TaZnSnO (0.5 mol%) | 1.26 ± 0.06 | 0.14 ± 0.005 | ~$10^5$ | 0.093 |
| TaZnSnO (1 mol%) | 2.29 ± 0.05 | 0.12 ± 0.050 | ~$10^5$ | 0.067 |
| TaZnSnO (2 mol%) | 3.43 ± 0.20 | 0.06 ± 0.004 | ~$10^4$ | 0.149 |

A positive bias stress (PBS) test was carried out to further study the effects of $Ta^{5+}$ doping on the stability of TaZnSnO TFTs, as shown in Figure 5a–d. The test conditions were under darkness with a stress voltage of 5 V at room temperature. The transfer curves of TaZnSnO TFTs both shifted toward the positive direction with the increase in stress time due to the electron trapping that occurred at the channel/dielectric interface. The method of extracting the electrical parameters in TFT can be found in Reference [36] and is summarized in Table 1. The value of $V_{th}$ increased from 0.29 to 3.43 V, and the $\mu$ decreased from 0.38 to 0.06 cm²/Vs with the $Ta^{5+}$ doping concentration increasing from 0 to 2 mol%. For oxide TFT, the $\mu$ is confirmed to be affected by both shallow traps near the conduction band and the generation of carriers from oxygen vacancies in the interface [29]. According to our results, the phenomenon of $\mu$ decreased with the increase in $Ta^{5+}$ concentration, indicating that $Ta^{5+}$ doping has a stronger effect on suppressing the number of electrons than making the electrons freer, which reduces the carrier concentration of ZnSnO films. In addition, with the $Ta^{5+}$ doping concentration increased from 0 to 1 mol%, the value of SS decreased from 0.103 to 0.067 V/decade. The SS is closely related to the total trap density ($N_{trap}$) [37], which can be extracted using the following equation:

$$N_{trap} = \left[ \frac{SS \log(e)}{kT/q} - 1 \right] \frac{C_i}{q} \tag{6}$$

where $C_i$, $T$, and $k$ are the capacitance per unit area of the insulator, the absolute temperature, and Boltzmann's constant, respectively. It should be noted that the *SS* increased from 0.067 to 0.149 V/decade, with the $Ta^{5+}$ doping rate continuing to increase to 2 mol%, which can be attributed to the generation of more trap states due to the higher doping concentration of $Ta^{5+}$.

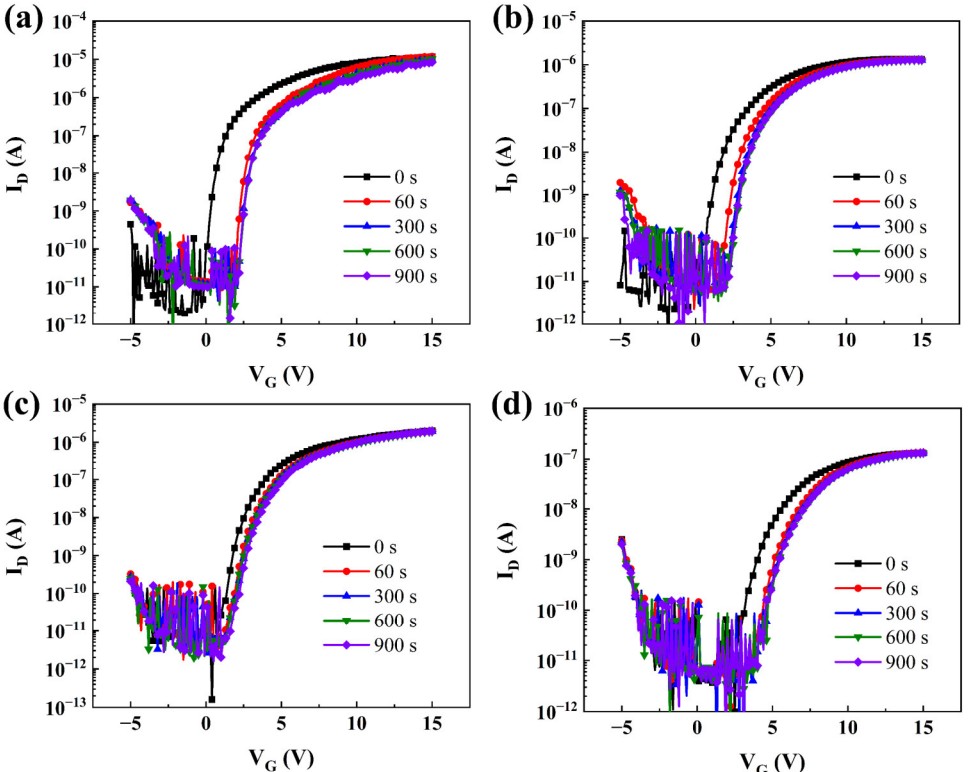

**Figure 5.** PBS of TaZnSnO TFTs with different $Ta^{5+}$ doping contents, (**a**) 0, (**b**) 0.5, (**c**) 1, and (**d**) 2 mol%.

Figure 6a demonstrates the threshold voltage shifts ($\Delta V_{th}$) of the transfer curves for the various TaZnSnO TFTs under the different PBS times. The undoped ZnSnO TFT shows a relatively large $\Delta V_{th}$ of 2.36 V. Compared with the ZnSnO TFT, the TaZnSnO TFTs show relatively small $\Delta V_{th}$ of 1.74, 0.71, and 1.57 V, and $Ta^{5+}$ doping concentrations of 0.5, 1, and 2 mol%, respectively. The results demonstrate that the PBS stability of TFTs was improved significantly with proper $Ta^{5+}$ content, as oxygen vacancies are suppressed by $Ta^{5+}$ doping. The time dependence of $\Delta V_{th}$ can be described by the stretching exponential equation [29]:

$$\Delta V_{th} = (V_G - V_{th}) \left\{ 1 - \exp\left[ -\left(\frac{t}{\tau}\right)^\beta \right] \right\} \tag{7}$$

where $\tau$ is the characteristics of carrier trapping time, and $\beta$ is the stretched-exponential exponent. The actual stress voltage applied to the device would be lower for higher $Ta^{5+}$-doped TFTs due to the increase in $V_{th}$. However, the characteristics of carrier trapping time $\tau$ and the stretched-exponential exponent $\beta$ hardly depend on the bias stress amplitude. According to the above formulas, the values of $\tau$ and $\beta$ extracted from the equation are $1.36 \times 10^5$, $1.62 \times 10^5$, $2.22 \times 10^6$, and $9.16 \times 10^4$ s, and 0.184, 0.397, 0.282, and 0.155 for the TaZnSnO TFTs with $Ta^{5+}$ doping of 0, 0.5, 1, and 2 mol%, respectively. The fitted curve described by Equation (7) is shown in Figure 6b. The $\tau$ is related to the potential barrier for the trapping layer, and an oxide TFT device with a high $\tau$ value demonstrates better stability because $\tau$ is related to the potential barrier for trapping as $\tau = \tau_0 \exp(E_\tau/kT)$ (where $E_\tau$ and $\tau_0$ are the average effective energy barrier and thermal prefactor for emission over the barrier, respectively). Accordingly, it is concluded that the 1 mol% $Ta^{5+}$-doped TaZnSnO TFT shows better stability due to lower trap density.

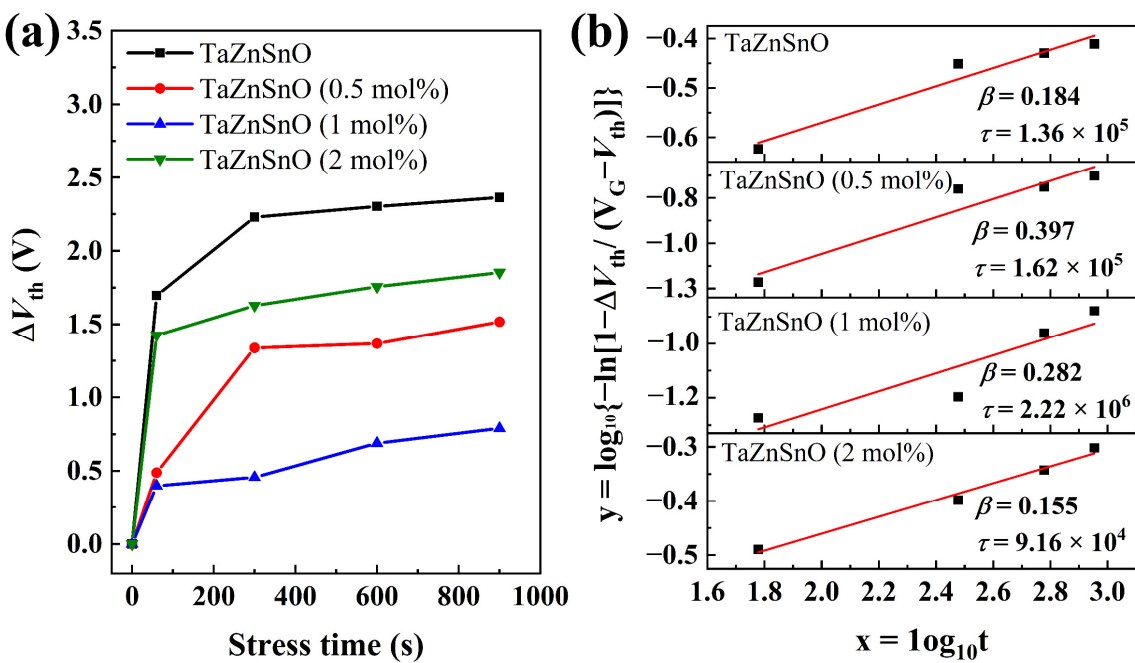

**Figure 6.** (**a**) The dependence of $\Delta V_{th}$ on stress time and (**b**) the fitted curves described by Equation (7) for the TaZnSnO TFTs with $Ta^{5+}$ doping of 0, 0.5, 1, and 2 mol%, respectively.

Although the TaZnSnO TFTs with 1 mol% $Ta^{5+}$ doping exhibited excellent stability, the electrical properties of TaZnSnO TFTs need to be further improved. Thus, the $Ta^{5+}$ doping method with the In:Zn:Sn ratio of 4:4:3 was carried out in Figure 7a. Compared with the TaZnSnO TFTs, the $\mu$ of TaInZnSnO TFT increases from 0.12 to 0.24 $cm^2/Vs$. The TaInZnSnO TFT shows a $\Delta V_{th}$ of 0.90 V, as shown in Figure 7b. In addition, the $V_{th}$ decreased from 2.29 to 0.76 V, the $I_{on}/I_{off}$ ratio increased from $10^5$ to $10^6$, and the *SS* increased from 0.067 to 0.167 V/decade, indicating the larger trapping at the channel-insulator interface, which leads to deterioration of the stability. The obtained $\tau$ and $\beta$ are $6.50 \times 10^6$ s and 0.23 for the TaInZnSnO TFT, respectively. Therefore, we conclude that a TaInZnSnO TFT with better stability and electrical performance can be obtained by doping $Ta^{5+}$ in an appropriate proportion.

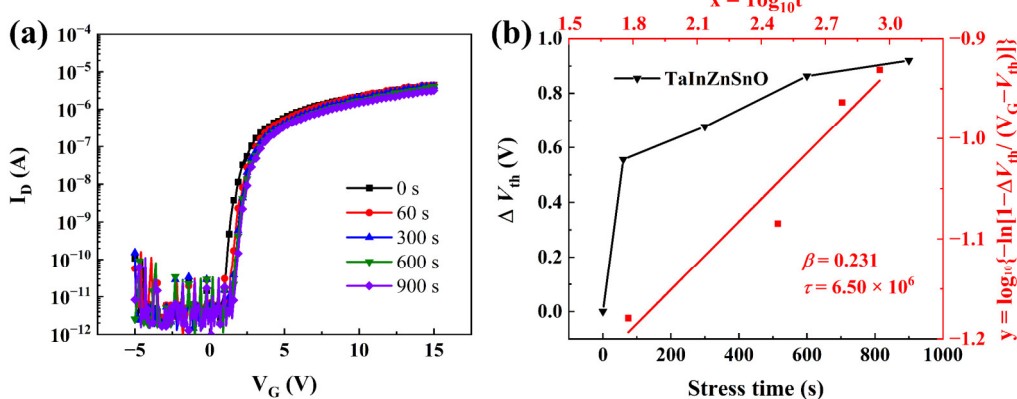

**Figure 7.** (**a**) PBS of TaInZnSnO TFTs. (**b**) The dependence of $\Delta V_{th}$ on stress time and the fitted curves described by Equation (7) for the TaInZnSnO TFT.

## 4. Conclusions

In summary, we first reported TaZnSnO TFTs fabricated by the solution method on ALD-$Al_2O_3$ films. The effects of different $Ta^{5+}$ doping concentrations on the microstructure were investigated, as well as the surface morphology and oxygen vacancy defect character-

istics of the films. The results indicate that UV–ozone treatment can significantly improve the hydrophilicity of $Al_2O_3$ films, which makes a better condition for the subsequent spin coating of the ZnSnO-based films. All of the films are amorphous, the RMS roughness of the films decreases slightly, and the optical transmittance of all samples shows about 90% transmission in the visible region. TaZnSnO TFTs with 1 mol% $Ta^{5+}$ doping have excellent stability relative to ZnSnO TFT, and its $\Delta V_{th}$ reduced from 2.36 to 0.71 V since the oxygen vacancy defects decreased from 38.05% to 18.70%. Additionally, $Ta^{5+}$-doped TaInZnSnO TFT with an In:Zn:Sn ratio of 4:3:3 was also developed to explore the electrical properties and stability of their associated TaInZnSnO TFTs. The results show that the $V_{th}$ of the TaInZnSnO TFT is relatively lower by a factor of three, and $\mu$ is twice as high as that of the TaZnSnO TFT while maintaining stability, with a $\Delta V_{th}$ of only 0.90 V. Consequently, such TaInZnSnO TFTs with highly stable and low $V_{th}$ characteristics can be applied in the field of high-resolution displays and sensors.

**Author Contributions:** Conceptualization: X.D.; data curation, X.D.; formal analysis, H.X., P.L., Z.C. and B.Y.; funding acquisition, X.D.; investigation, B.W. and C.F.; methodology, H.X., B.W. and C.F.; project administration, X.D. and J.Z.; resources, X.D.; software, H.X., P.L., Z.C., B.Y. and C.F.; supervision, X.D. and J.Z.; validation, P.L., B.W., C.F. and X.D.; writing—original draft, H.X. and X.D.; writing—review and editing, H.X. and X.D. All authors have read and agreed to the published version of the manuscript.

**Funding:** This work is supported by the National Natural Science Foundation of China (62274105), and C. Fu acknowledges the support of the National Natural Science Foundation of China (21902063).

**Institutional Review Board Statement:** Not applicable.

**Informed Consent Statement:** Not applicable.

**Data Availability Statement:** Not applicable.

**Conflicts of Interest:** The authors declare no conflict of interest.

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
