# Peer review of "Enhanced Stability of Solution-Processed Indium–Zinc–Tin–Oxide Transistors by Tantalum Cation Doping"

_coatings, doi:10.3390/coatings13040767_

Round 1
Reviewer 1 Report
After reading, I believe that manuscript should be modified. See below points.
1- the Al2O3 gate dielectrics of approximately 50 nm thick were formed 88 on the heavily doped p-type Si substrates. Write if substrate is Si(100) or Si(111). There is big different between Si(100) and Si(111).
2- What does it mean (The prepared devices were annealed at 300 °C in the air for 98 15 min on the hot annealing furnace.)?
3- In "Figure 3. XPS survey spectra of the TaZnSnO thin films with different Ta5+ doping.:, where is OKLL, Auger peak. This spectrum should be deconvoluted.
4- Threshold voltage, carrier mobility, ION/IOFF in Table 1 is a good result for ZnSnO, but I could not understand why author add Ta? I suggest addressed some references such as {1-Applied Physics A 125, 1-7. 2019. 2- Journal of Electronic Materials 47, 3717-3726, 2018. 3- Current Applied Physics 18 (12), 1546-1552, 2018 } to confirm this claim why lower these parameters are OK.
5- Eq (5) needs a reference.
6- With Williamson- Hall equation or relation, is it possible to get more details about Stress- strain with looking at XRD patterns. Is it true?
7- The results show that 271 the Vth of the TaInZnSnO TFT is relatively lower by a factor of three and μ is twice as high 272 as that of the TaZnSnO TFT, while maintaining stability with a ΔVth of only 0.90V. It is so good result. But without Ta, results seem better. Explain it.
Reviewer 2 Report
The article "Enhanced Stability of Solution-Processed Indium-Zinc-Tin-Oxide Transistors by Tantalum Cation Doping" presents the results of studies on the synthesis and subsequent doping of thin-film transistors based on zinc-tin oxide (ZnSnO). In general, this line of research has great prospects not only in fundamental terms, but also has great potential for further practical use. However, despite the fact that the authors have made a number of changes to the text of the article, they should give answers to the following questions of the reviewer.
1. The authors should reflect how exactly they measured the value of the contact wetting angle of the obtained coatings.
2. The authors should show how exactly the roughness and waviness of the films changed depending on the production conditions?
3. The presented optical transmission spectra should be presented in a more presentable form, as well as an explanation of how exactly the transmission values for the obtained films were changed.
4. The results of X-ray diffraction, reflecting changes in the amorphous nature of the obtained films, are rather little informative. Should the authors explain small reflexes and their presence?
5. For all observed values, authors should provide measurement errors as well as standard deviation values.
6. Also in the conclusion, the authors should reflect the new data, taking into account the comments of the reviewer, as well as reflect data on further research prospects.
Reviewer 3 Report
Referee report on manuscript “Enhanced Stability of Solution-Processed Indium– Zinc–Tin Oxide Transistors by Tantalum Cation Doping”
This is a fairly good and necessary article, and it can be recommended for publication, but only after clarification / improvement of some ambiguities.
1. Introduction. The comment about oxygen vacancies is important, but the situation is even more complicated because the vacancies are in several charge states. This is known for both binary and complex oxides. See, for example, review paper: Popov, A. I., Kotomin, E. A., & Maier, J. (2010). Basic properties of the F-type centers in halides, oxides and perovskites. Nuclear Instruments and Methods in Physics Research Section B: Beam Interactions with Materials and Atoms, 268(19), 3084-3089.
2. Note that diffusion characteristics of vacancies depend on their charge state.
Line 159 -165. This conclusion looks unjustified, since such low concentrations cannot change the band width, but lead to the appearance of impurity-related electronic states (levels) at the optical absorption edge. See the latest article by the Editors of “Optical Materials” journal: Brik, M. G., Srivastava, A. M. (2022). A few common misconceptions in the interpretation of experimental spectroscopic data. Optical Materials, 127, 112276. The ambiguity of drawing straight lines (Fig. 2d) was specifically discussed in this work
3. In this regard, the question arises, how was the defectiveness of the obtained films (luminescence, Raman) checked?
4. More information on porosity of the obtained films would be also helpful.
5. The first 10 references are quite old, and therefore the question arises how relevant this study is and whether the authors are familiar with the latest (last 5 years) achievements.
6. In the conclusions, it is necessary to clearly formulate what new data about the studied material were obtained in this work?
In general, the manuscript is interesting and can be considered for publication after constructive reflection on the above comments.
Round 2
Reviewer 2 Report
The authors answered all the questions, the article can be accepted for publication.
Reviewer 3 Report
The authors have successfully improved the original version of the manuscript, responding constructively to all the comments/recommendations of the reviewer. The article can be recommended for publication.